# RAG in the Aerospace Domain: Retrieval, Generation, and User Evaluation for NASA Documentation

Dominykas Petniunas[1], Gabriel Iturra-Bocaz[2], and Petra Galuščáková[3]

[1,2,3]University of Stavanger
{d.petniunas, gabriel.e.iturrabocaz, petra.galuscakova}@uis.no

## Abstract

Large Language Models (LLMs) have demonstrated remarkable capabilities in Natural Language Understanding and text generation, but their application is often limited by hallucinations, outdated knowledge, and lack of evidence. Retrieval-Augmented Generation (RAG) addresses these fundamental LLM limitations by integrating external knowledge sources, thereby improving the factual accuracy and traceability while maintaining the text generative capabilities. This work presents the design and implementation of a web-based RAG system for the aerospace domain, leveraging more than 10,000 NASA technical documents and lessons-learned mission reports. The system integrates open-source LLaMA and closed-source OpenAI models and performs an extensive comparative analysis of their performance within the RAG framework. Evaluation through both automated metrics and user studies demonstrates the effectiveness of the RAG approach for both technical and non-technical users. The findings provide insights and establish a foundation for future advancements in AI-driven knowledge management for specialized fields[1].

## 1 Introduction

Large organizations face a critical challenge: leveraging years or decades of accumulated knowledge and lessons learned to inform new projects and decisions. This problem is particularly acute in high-risk, high-stakes environments. An example of such an environment is NASA, an organization with thousands of past projects that span decades. NASA engineers who work on new missions often struggle to find relevant historical information that could prevent costly mistakes or accelerate innovation. The core challenge lies in the complexity of discovering knowledge within vast and diverse available collections. NASA, as a large and long-established organization that works with multiple contractors, faces several barriers: vocabulary differences across time periods, varying terminology between contracting companies, and the sheer scale of documentation. This creates a scenario where critical lessons learned, such as the infamous O-ring failure that led to the Challenger disaster in 1986, may be documented, but remain inaccessible to engineers working on similar components in new projects [1]. The result is a knowledge gap where valuable insights from past missions remain isolated and unused. This challenge is exemplified by NASA's efforts to develop risk digital assistants that can extract and leverage past project data for predictive decision-making[2].

This problem extends beyond aerospace to any large organization with extensive historical documentation: healthcare systems with decades of patient data, legal firms with case histories, or manufacturing companies with safety records. In these domains, the ability to quickly and accurately retrieve relevant historical information can significantly impact decision quality, risk assessment, and project outcomes.

RAG has emerged as a leading approach that enhances LLMs by seamlessly integrating external knowledge sources with text generation capabilities [2]. This hybrid approach allows systems to take advantage of both the generative strengths of LLMs and the precision of Information Retrieval (IR) techniques, producing responses drawn from a substantially broader knowledge base than what is encoded in the model parameters alone. While RAG systems have demonstrated effectiveness across various domains, the specific challenges of applying them to highly specialized technical documentation in aerospace environments present unique opportunities for research.

The core contribution of this work is addressing this knowledge discovery challenge through a specialized RAG system designed for NASA's documentation. By processing more than 10,000 NASA technical documents [3] and lessons-learned mission reports [4], this system allows engineers to quickly access verified information with direct source references, significantly reducing the time and effort required to find relevant historical data. The system integrates and compares open-source LLaMA and closed-source OpenAI models, addressing the challenge of making NASA's extensive technical docu-

---

[1]Code and instructions for preparing the datasets can be found at https://github.com/DominykasPe/Master_Thesis_RAG

[2]https://techport.nasa.gov/projects/117547

Proceedings of the 7th Northern Lights Deep Learning Conference (NLDL), PMLR 307, 2026.

mentation and lessons-learned databases more accessible and trustworthy. Beyond the proof-of-concept application, this work contributes a publicly available data collection pipeline and evaluation methodology combining automated metrics with user studies, providing a foundation for future research in domain-specific RAG applications.

This paper is organized as follows: next section reviews existing literature in the domains of RAG, conversational agents, and evaluation methodologies, focusing on applications in specialized domains and space sector implementations. Our approach is described in Section 3 and the evaluation is discussed in Section 4. We conclude the paper in Section 5.

## 2 Related Work

The RAG architecture was first introduced by Lewis et al. [5], combining a retrieval component with a sequence-to-sequence generative model to enhance text generation with external knowledge. Unlike traditional LLMs that rely solely on their pre-trained knowledge, RAG allows models to generate high-quality text supported by relevant external information, making it particularly effective for knowledge-intensive tasks [5]. The effectiveness of retrieval-augmented approaches has been demonstrated in open-domain Question Answering (QA) systems [6, 7] and dialogue systems [8], where the integration of retrieval mechanisms has improved contextual understanding and response generation capabilities.

**Advanced RAG Architectures.** While this work implements a vanilla RAG architecture with dense retrieval and direct generation, research has explored more advanced designs to enhance retrieval quality and generation accuracy.

Advanced approaches include multi-stage retrieval combining sparse methods (BM25) with dense retrievers [9], re-ranking models (monoT5 [10], monoELECTRA [11]) that refine results using cross-encoders, and graph-based methods leveraging document relationships and Abstract Meaning Representation (AMR) graphs [12]. These methods address limitations in vanilla RAG by incorporating structural information: knowledge graph-based systems [13] integrate external knowledge bases for improved answer extraction, while document graph approaches [13] establish connections between retrieved documents to identify weakly connected relevant information.

**Conversational Agents in Space Sector.** The development of conversational QA systems has advanced significantly with the rise of LLMs, enabling models to answer questions based on given contexts, often involving RAG when documents surpass the language model's context window [14]. Studies focus primarily on text-only documents (such as regulations, manuals, and technical reports) from various domains [15–17] and documents combining plain text with tables (such as Wikipedia articles with tabular data, financial reports, and semi-structured knowledge sources) [18–20].

Several projects have explored the use of AI-powered virtual assistants to aid engineers during spacecraft mission design. The most notable ones are Daphne [21], the Design Engineering Assistant (DEA) [22], and SpaceQA [23]. DEA is a non-intrusive decision support tool that enhances expert perception of different design alternatives and past decision outcomes through Natural Language Processing, Machine Learning, Knowledge Management, and Human-Machine Interaction methods [22]. Daphne, evaluated at NASA's Jet Propulsion Laboratory (JPL), enhances design task performance through a microservices architecture featuring a web front-end, server (Daphne Brain), and software roles that interface with structured knowledge graphs for better design inputs [24]. SpaceQA, developed by the European Space Agency (ESA), is an open-domain QA system for space mission design, employing a dense retriever and neural reader similar to an RAG pipeline [24].

**LLM-as-a-Judge Evaluation in RAG Systems.** Traditional evaluation of the RAG systems require manually created ground truth data, which are typically quite expensive to acquire. Traditional metrics like ROUGE and BLEU fail to capture the nuanced quality dimensions required for RAG evaluation, particularly factual correctness and contextual grounding. New evaluation methods, which employ LLMs, thus recently started to emerge in RAG evaluation. Wang et al. [25] demonstrate that ChatGPT can effectively evaluate text generation by providing scores on 0-100 or 1-5 star scales for aspects such as relevance, factual accuracy, and groundedness, achieving state-of-the-art correlation with human judgments across multiple NLG tasks. However, this approach exhibits sensitivity to prompt design and reduced effectiveness on datasets with strong lexical biases.

Muhamed [26] introduces the CCRS (Contextual Coherence and Relevance Score), employing LLaMA-70B as a zero-shot judge to evaluate RAG systems in five dimensions: contextual coherence, question relevance, information density, answer correctness, and information recall. Their evaluation on the BioASQ biomedical dataset shows promising results across multiple RAG configurations with different readers and retrievers, while highlighting the challenge of achieving perfect factual accuracy in complex biomedical domains.

Building on these evaluation foundations, broader investigations have explored the reliability and effectiveness of LLM judges across different contexts. Tseng et al. [27] conduct the first systematic evaluation of LLMs as expert-level data annotators across

finance, biomedicine, and law domains, finding that models average 35% behind human expert performance despite showing promise in general NLP tasks. Ashktorab et al. [28] develop EvalAssist, comparing direct assessment versus pairwise comparison strategies, and demonstrate that practitioners prefer direct assessment for clarity while using pairwise comparison for subjective evaluations. Bavaresco et al. [29] present JUDGE-BENCH, a comprehensive benchmark evaluating 11 LLMs across 20 datasets, revealing substantial variance in model performance and better alignment with non-expert versus expert human judges. Thus, although the LLM-as-a-Judge Evaluation still suffers from multiple issues, due to its low cost, it is now a standard for RAG evaluation, often accompanying expensive user studies.

**User Evaluation Studies in RAG Systems.** While technical metrics provide important insights into RAG performance, user-centered evaluation remains critical for understanding real-world effectiveness and adoption. Hasan et al. [30] present a comprehensive study of five domain-specific RAG applications deployed across governance, cybersecurity, agriculture, industrial research, and medical diagnostics. Their approach combined Likert-scale surveys with open-ended qualitative feedback to capture both measurable insights and descriptive user experiences, ultimately documenting key lessons learned from user feedback to guide future RAG system development and deployment practices.

## 3  Method

### 3.1  RAG Framework

RAG is a hybrid approach that combines two essential components: (i) a retrieval system that pulls documents from an external knowledge base, and (ii) a generation component that uses this information to create natural, human-like text [31, 32]. By blending these capabilities, RAG models can produce coherent and fluent responses while anchoring their output in current real-world information. The workflow is illustrated in Figure 1.

The generation component leverages the contextual information to produce responses that maintain the natural language capabilities of LLMs while being factually anchored. The system can be configured to prioritize retrieved context over pre-trained knowledge, ensuring that responses are based primarily on the retrieved NASA documents rather than the model's training data [33].

### 3.2  RAG System Implementation

The core pipeline consists of six key steps: data gathering, chunking, vectorization, storage, retrieval, and response generation.

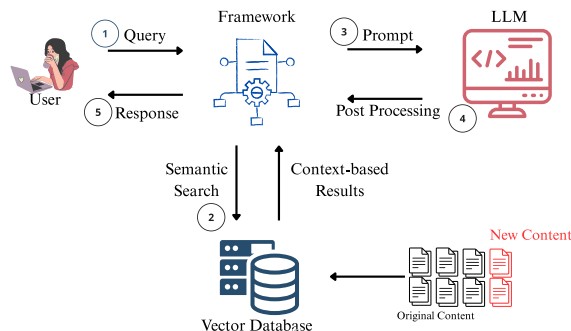

**Figure 1.** RAG Architecture workflow: (1) User submits query to the system, (2) Query is vectorized and semantic search is performed against the document vector database, (3) Top-k relevant documents are retrieved and passed as context, (4) LLM generates response using both the original query and retrieved context, and (5) presents it to the user.

### 3.2.1  Data Gathering

To be able to use the data, which would support our use case of vast knowledge of a single organization, we decided to create a novel collection consisting of freely available NASA technical resources. We used web scraping techniques to collect information from NASA Technical Report Server[3], which contains papers, patents, reports and other technical materials created or funded by NASA. The server contains thousands of documents, collected since 1915, though our system focuses on documents from the 2000s onwards to ensure relevance to current aerospace practices. In addition to this, we used documents available at the NASA Lessons Learned website[4] which contains reviewed lessons learned from NASA programs and projects. Together, we collected 1,859 records from the Lessons Learned database, and 8,143 PDF files from the NASA Technical Report Server. The collection contains various aerospace-related materials, on topics such as spacecraft design, propulsion systems, and mission operations.

The Selenium library was employed to automate browser interactions and handle JavaScript-rendered content, while BeautifulSoup was utilized for parsing HTML and navigating through hundreds of pages efficiently. This combination was particularly needed for NASA's websites, which required navigating through complex page hierarchies and extracting data from dynamically loaded content. The web scraping process required careful pacing to avoid triggering rate-limiting mechanisms, with individual PDF files often exceeding 30MB due to extensive technical content and images.

---

[3]https://ntrs.nasa.gov/search
[4]https://llis.nasa.gov/

### 3.2.2 Document Processing and Chunking

**Chunking Strategy Analysis.** Three primary chunking approaches were evaluated for this domain-specific application: (i) *recursive chunking*, which splits text based on hierarchical separators (paragraphs, sentences, words), (ii) *semantic chunking*, which groups text based on semantic similarity using embedding models to identify natural breakpoints, and (iii) *section-based chunking*, which leverages document structure to maintain semantic boundaries.

Recursive chunking, while computationally efficient, was unsuitable for NASA documentation as it often split technical procedures across multiple chunks. Semantic chunking required significantly higher processing power than section chunking, and the chunks were too large to be useful for a system with limited context window.

**Section-Based Chunking Implementation.** The implemented approach utilizes section-based chunking through the *pymupdf4llm*[5] library, which employs a multi-stage algorithm specifically designed to extract PDF content in formats optimized for LLM and RAG environments. The library's algorithm first converts PDF pages to GitHub-compatible markdown format while preserving document structure through font analysis, header detection, and formatting preservation. Subsequently, it identifies section boundaries using font size variations, header patterns (both # markdown syntax and bold formatting), and hierarchical document structure.

For a document $D$ with $n$ sections, the chunking function $\phi : D \rightarrow C$ produces chunks $C = \{c_1, c_2, \ldots, c_n\}$ where each chunk $c_i$ maintains semantic coherence and includes metadata (chunk ID, title, content, page number, section number, source URL, and statistics) enabling full traceability to original NASA materials.

**Dataset Structure Analysis.** As illustrated in Figure 2, most Technical Reports contain between 5 and 20 chunks, as the NASA documents vary widely from brief single-page reports to extensive technical manuals exceeding 50 pages[6]. This distribution shows consistent chunking patterns across the collection. The figure shows a distribution capped at 80 chunks per document. We opted for this to limit the processing of the outlier PDFs with complex tables and figures.

### 3.2.3 Vectorization and Retrieval

For vectorization, the all-MiniLM-L6-v2[7] transformer model is utilized, which is a lightweight but

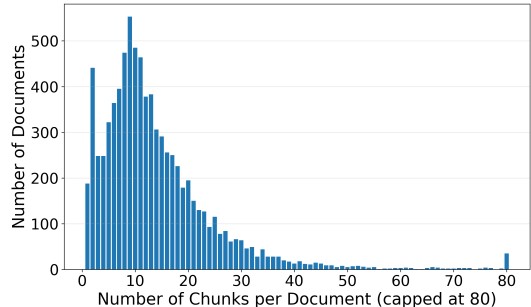

**Figure 2.** Distribution of NASA Technical Reports by chunk count showing the frequency of documents containing different numbers of chunks. Documents are capped at 80 chunks per document due to processing limitations that come from complex table formatting and images in certain PDFs.

effective embedding model designed for semantic search tasks. The vectors are stored in a FAISS[8] database for efficient similarity search. For a user query $q$, its embedding $\mathbf{v}_q$ is computed and the top-$k$ most similar chunks are retrieved using cosine similarity. The retrieval process returns chunks $R_k = \{c_{r_1}, c_{r_2}, \ldots, c_{r_k}\}$ where $r_j$ represents the rank of chunk $c_{r_j}$ based on similarity scores. Retrieval quality is evaluated using standard metrics including precision, recall, Mean Reciprocal Rank (MRR), Hit Rate, and Normalized Discounted Cumulative Gain (NDCG). The impact of different retrieval configurations (varying k values and cosine similarity thresholds) is examined in the Results section (Table 2).

### 3.2.4 Language Model Integration

The system integrates both open-source and closed-source language models to enable a comprehensive comparison between paid and freely available options. The GPT models (GPT-4o-mini and GPT-3.5-turbo) were selected for their cost-effectiveness while maintaining strong performance capabilities, making them accessible for budget-focused research applications. For open-source alternatives, LLaMA-3.3-70B-Instruct-Turbo and LLaMA-3.2-1B-Vision-Instruct represent the only freely available models on Together.ai's platform[9], ensuring consistent cloud-based resource allocation critical for fair model comparison.

The system uses *ChatPromptTemplate*[10] from LangChain, a library designed for creating structured and concise prompt instructions. This template formats the user's question with the retrieved context, creating a structured input that guides the

---

[5]https://pypi.org/project/pymupdf4llm/

[6]The Lessons Learned are typically single-page reports that convert entirely into individual chunks and are thus not included in the distribution.

[7]https://huggingface.co/sentence-transformers/all-MiniLM-L6-v2

[8]https://faiss.ai/index.html

[9]https://www.together.ai/models

[10]https://python.langchain.com/api_reference/core/prompts/langchain_core.prompts.chat.ChatPromptTemplate.html

language model to provide accurate, contextually grounded answers based on the NASA documentation rather than relying solely on its pre-trained knowledge.

For evaluation purposes, all four models are assessed in the automated generation evaluation to provide a comprehensive performance comparison. However, given the limited participant count in the user study, only GPT-4o-mini and LLaMA-3.3-70B-Instruct-Turbo are included in the user evaluation to ensure sufficient statistical power while representing both commercial and open-source model categories.

## 3.3 Evaluation Framework

The evaluation approach is built upon a our constructed test dataset consisting of 50 records that are used for both retrieval and generation evaluation. Each record contains a manually crafted question created from randomly selected document chunks, paired with expected answers derived from the chunk content. For answer preparation, either the complete chunk content was used or GPT-4 was employed to extract only the parts relevant to the specific problem, ensuring that evaluation answers remained strictly grounded in the original NASA documentation while avoiding potential human error in test dataset creation. The expected answers are standardized to 150-200 tokens in length, with each record referencing 1-3 relevant document chunks. Table 1 illustrates the structure of a typical evaluation record.

**Table 1.** Example evaluation record from the test dataset.

| Question | What issue was discovered with the ILT radiator during the CALIPSO satellite-level thermal vacuum test? |
| --- | --- |
| Expected Answer | During the satellite-level thermal vacuum test of CALIPSO, which aimed to demonstrate positive thermal control of all payload components within their required temperature limits under... |
| Relevant Chunks | 20070021525.pdf_11, 20070021525.pdf_12 |

### 3.3.1 Automated Evaluation

Generation quality is evaluated using a multifaceted approach that uses ROUGE scores, BERTScore for estimating semantic similarity, and LLM-as-a-Judge evaluation.

BERTScore computes semantic similarity by summing cosine similarities between token embeddings. For reference sentence $x$ and predicted sentence $y$, the F1 measure is calculated as:

$$F_{BERT} = 2\frac{P \times R}{P + R} \qquad (1)$$

where $P$ and $R$ represent precision and recall based on maximum cosine similarities between tokens of the generated and ground truth answer [34].

ROUGE metrics evaluate n-gram overlap between generated and reference texts. The ROUGE-N score is calculated as:

$$\text{ROUGE-N} = \frac{\sum_{S \in \mathcal{R}} \sum_{g \in S} \text{Count}_{match}(g)}{\sum_{S \in \mathcal{R}} \sum_{g \in S} \text{Count}(g)} \qquad (2)$$

where $\mathcal{R}$ represents reference summaries, $g$ represents n-grams, and $\text{Count}_{match}(g)$ is the maximum number of matching n-grams of the generated and ground truth answer [35].

### 3.3.2 LLM-as-a-Judge Methodology

This work utilizes two LLM-as-a-Judge methods: the first uses OpenAI's GPT-4o-mini model for evaluation, while the second implements G-Eval [36], a framework that uses a chain-of-thought (CoT) approach to evaluate the quality of generated text through structured evaluation forms. The main LLM-as-a-Judge scores are *Correctness*, *Relevance*, *Accuracy*, and *Groundedness*. These dimensions are particularly critical for NASA applications: Correctness ensures that factual claims match documented mission outcomes, Relevance verifies that responses address the specific engineering question posed, Accuracy validates that technical specifications align with source documentation, and Groundedness confirms traceability to authoritative NASA sources rather than hallucinated content. For end users, groundedness scores above 0.85 indicate high confidence that retrieved information originates from verified NASA documentation, addressing the knowledge gap where engineers previously struggled to trace historical decisions back to their sources.

The custom GPT-4o-mini evaluation uses a single prompt[11] requesting scores on a 0-10 scale for three criteria (relevance, factual accuracy, groundedness) with direct numerical output. This design choice, while computationally efficient, introduces potential conflation of evaluation criteria: the model may struggle to independently assess whether an answer is "relevant to the question" versus "factually accurate according to source documents." For instance, when evaluating a response about spacecraft thermal systems, the single-prompt judge might rate factual accuracy highly simply because the answer discusses thermal management, without carefully checking if specific temperature values match the source documents.

In contrast, G-Eval employs a more structured CoT approach that decomposes evaluation into sequential reasoning steps for four criteria (correctness, relevance, accuracy, groundedness), forcing the model to independently assess each dimension. This prevents the model from conflating evaluation dimensions and applies stricter standards when assessing technical precision. This dual approach enables evaluation using both single-prompt and multi-prompt

---

[11]GPT-4o-mini uses custom ChatPromptTemplate prompts.

methodologies for comprehensive RAG system assessment.

### 3.3.3 User Study Design

The user study was conducted with 20 participants from diverse non-technical backgrounds, with only 3 participants having IT experience, ensuring evaluation from the general audience perspective. Each session lasted 15-25 minutes, during which each participant was asked to complete five distinct engineering tasks designed to simulate realistic scenarios requiring specialized aerospace knowledge retrieval.

**Task Design and Implementation.** The five tasks covered critical aerospace domains: (1) Engine Rollback Investigation, requiring analysis of Propulsion Systems Laboratory data on ice particle effects; (2) Satellite Thermal System Evaluation, focusing on CALIPSO payload thermal performance; (3) Aircraft Noise Profile Assessment, examining noise components across different flight conditions; (4) Critical Power System Design, investigating Uninterruptible Power Supply applications; and (5) Electronics System Safety Review, identifying analytical methods for circuit problem detection. Tasks were created by randomly choosing document chunks, manually creating problems from those chunks. GPT model was only used for acquiring supplementary information such as creating scenarios to assist users in understanding the problem. Each task required participants to find at least three of seven predefined keywords to demonstrate successful knowledge acquisition. Figure 3 illustrates the structure and complexity of a typical user study task.

**Task 1: Engine Rollback Investigation**

You're an aerospace engineer analyzing engine performance in icing conditions. Your team needs to understand what particle characteristics lead to engine rollback events. Research the Propulsion Systems Laboratory (PSL) test data findings to determine critical ice particle sizes and temperature conditions that contribute to these events.
Expected findings should include:

- Analysis of Propulsion Systems Laboratory (PSL) data points on the LF11 engine model
- Critical particle size requirements for engine rollback (when thrust unexpectedly decreases)
- Relevant wet bulb temperature range in the Low Pressure Compressor (LPC) region

**Figure 3.** Example of Task 1 in User Evaluation Study: Engine Rollback Investigation interface showing the structured engineering scenario presented to participants.

**Study Protocol and Bias Mitigation.** Participants received authentication credentials and accessed a web application that provided clear instructions and system descriptions. To prevent cheating and ensure authentic interaction with the RAG system, copy functionality was disabled for task descriptions, requiring users to reformulate queries in their own words. The system randomly assigned different models (LLaMA-3.3-70B-Instruct-Turbo or GPT-4o-mini) to each task per user, ensuring balanced model comparison. Upon completion, participants provided feedback through the System Usability Scale[12] (SUS) questionnaire supplemented with custom questions addressing task difficulty, AI assistant helpfulness, and system improvement suggestions.

## 4 Results

**Retrieval Performance.** To evaluate the impact of retrieval parameters on system performance, we conducted extensive experiments across different configurations of document count (k) and cosine similarity thresholds (minimum similarity between query and document embeddings required for retrieval), as shown in Table 2. The system was tested across 50 questions, where each question has one or several expected answer chunks from potentially different source files.

The results reveal important trade-offs between precision and recall across different configurations. The baseline configuration (k=3, threshold=None), which was used in subsequent generation and user evaluation experiments, follows standard RAG practice for balancing context richness with computational efficiency. This configuration achieves a recall of 0.66 and hit rate of 0.68, finding the majority of expected documents and locating at least one relevant document for most queries. The MRR of 0.64 indicates that relevant documents consistently appear in top positions.

Notably, the k=3 configuration with threshold=0.8 achieves the highest precision (0.39) while maintaining strong recall (0.62), suggesting this might represent an optimal configuration for applications prioritizing answer accuracy. Conversely, higher k values with no threshold filtering (e.g., k=10, threshold=None) maximize recall (0.69) and NDCG (0.65) at the cost of lower precision (0.07). Lower threshold values (0.5-0.7) showed poor performance and were thus only evaluated for k=3. While the precision scores appear modest, this may reflect the system's ability to find additional relevant documents beyond those manually marked in the test dataset.

**Generation Capabilities.** The generation evaluation demonstrates that the RAG system success-

---

[12]https://www.surveylab.com/blog/system-usability-scale-sus/

**Table 2.** Retrieval performance comparison across different numbers of retrieved documents (k) and cosine similarity thresholds (minimum similarity required for document inclusion). Each configuration was tested on 50 questions. Threshold=None indicates no filtering.

| Documents Retrieved | Similarity | Prec. | Rec. | MRR | Hit Rate | NDCG |
|---|---|---|---|---|---|---|
| 3 | None | 0.23 | 0.66 | 0.64 | 0.68 | 0.64 |
| 3 | 0.5 | 0.08 | 0.10 | 0.10 | 0.10 | 0.10 |
| 3 | 0.6 | 0.28 | 0.33 | 0.33 | 0.34 | 0.33 |
| 3 | 0.7 | 0.38 | 0.51 | 0.49 | 0.52 | 0.49 |
| 3 | 0.8 | **0.39** | 0.62 | 0.60 | 0.64 | 0.60 |
| 3 | 0.9 | 0.32 | 0.64 | 0.62 | 0.66 | 0.62 |
| 5 | None | 0.14 | 0.68 | 0.64 | **0.70** | 0.65 |
| 5 | 0.8 | 0.36 | 0.62 | 0.60 | 0.64 | 0.60 |
| 5 | 0.9 | 0.28 | 0.66 | 0.62 | 0.68 | 0.63 |
| 7 | None | 0.10 | 0.68 | 0.64 | **0.70** | 0.65 |
| 7 | 0.8 | 0.36 | 0.62 | 0.60 | 0.64 | 0.60 |
| 7 | 0.9 | 0.26 | 0.66 | 0.62 | 0.68 | 0.63 |
| 10 | None | 0.07 | **0.69** | 0.64 | **0.70** | **0.65** |
| 10 | 0.8 | 0.35 | 0.62 | 0.60 | 0.64 | 0.60 |
| 10 | 0.9 | 0.25 | 0.67 | 0.62 | 0.68 | 0.63 |

fully addresses the core challenge of providing accurate, contextually grounded responses from NASA documentation. Table 3 presents the results using only queries for which at least one relevant document was retrieved (34 of 50 questions), corresponding to the 68% hit rate. Performance across all four models is remarkably consistent, underscoring the critical role of high-quality retrieval in RAG systems.

The results show strong semantic performance with BERTScore F1 values between 0.78-0.80, indicating that generated answers are semantically very close to ground truth. For NASA engineers, BERTScore values above 0.75 signal that the system preserves technical meaning even when using different wording, while ROUGE-1 scores of 0.46-0.48 demonstrate strong unigram overlap with approximately half of the technical terms from reference answers appearing in generated responses.

The comparative analysis between custom LLM-as-a-Judge and G-Eval metrics reveals important methodological insights. While the custom GPT-4o-mini approach yields higher factual accuracy scores (0.88-0.90) compared to G-Eval accuracy scores (0.81-0.85), the remarkably high G-Eval relevance scores (0.96-0.98) compared to custom relevance scores (0.84-0.87) demonstrate that focused single-criterion prompts enable more precise assessment. The lower G-Eval accuracy scores suggest that multi-prompt evaluation applies stricter standards when assessing technical precision, better identifying cases where responses use approximately correct aerospace terminology without precisely matching source specifications.

The high groundedness scores (0.87-0.88) confirm that the system successfully leverages NASA documentation rather than relying on pre-trained

knowledge, ensuring factual accuracy and traceability. The consistency across models suggests that retrieval quality, rather than model choice, determines RAG performance for specialized aerospace documentation.

**User Evaluation Results.** The user evaluation provides crucial validation that the RAG system successfully addresses the challenge of knowledge discovery from the introduction. Figure 4 shows the completion times of tasks in different models, while Figure 5 shows the number of attempts by users to complete various tasks.

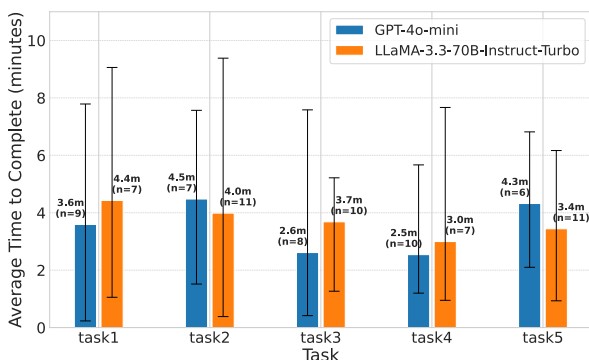

**Figure 4.** Comparison of GPT-4o-mini and LLaMA-3.3-70B-Instruct-Turbo models showing average task completion times in minutes. Each bar represents the mean completion time, with n indicating the number of users who completed each task. Vertical black lines show the minimum and maximum completion times for each model and task combination.

Most notably, the low attempt counts reveal that users with no aerospace domain experience successfully solved relatively complex aerospace engineering scenarios in just a few tries. The first three tasks

**Table 3.** Generation evaluation results on the retrieved documents (34 questions).

| Metric | GPT4o-mini | GPT3.5-turbo | LLaMA-70b | LLaMA-vision |
|---|---|---|---|---|
| *Semantic Metrics* | | | | |
| BERTScore Precision | 0.74 | **0.78** | 0.75 | 0.75 |
| BERTScore Recall | **0.83** | 0.82 | **0.83** | **0.83** |
| BERTScore F1 | 0.78 | **0.80** | 0.79 | 0.78 |
| Semantic Similarity | **0.85** | **0.85** | **0.85** | 0.82 |
| *ROUGE Metrics* | | | | |
| ROUGE-1 | 0.46 | **0.48** | 0.46 | 0.47 |
| ROUGE-2 | 0.26 | **0.29** | 0.26 | **0.29** |
| ROUGE-L | 0.45 | **0.48** | 0.45 | 0.47 |
| *LLM-as-a-judge Metrics* | | | | |
| Answer Relevance | 0.86 | 0.85 | **0.87** | 0.84 |
| Factual Accuracy | 0.89 | 0.88 | **0.90** | 0.89 |
| Groundedness | 0.88 | 0.87 | **0.88** | **0.88** |
| *G-Eval Metrics* | | | | |
| G-Eval Correctness | **0.79** | 0.75 | **0.79** | 0.74 |
| G-Eval Relevance | **0.98** | 0.97 | 0.97 | 0.96 |
| G-Eval Accuracy | **0.85** | 0.81 | 0.84 | 0.81 |
| G-Eval Groundedness | **0.87** | 0.86 | **0.87** | **0.87** |

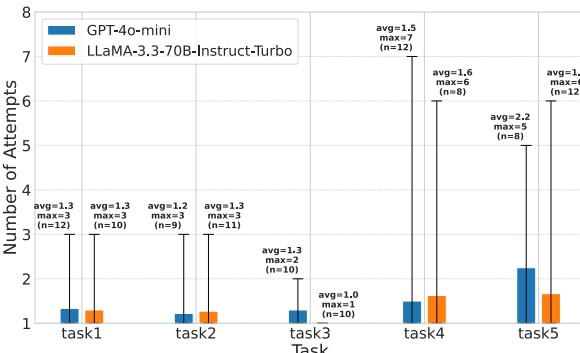

**Figure 5.** Comparison of GPT-4o-mini and LLaMA-3.3-70B-Instruct-Turbo models showing the average number of attempts users required to complete each task. Each bar displays the mean number of attempts, with n indicating the number of users who completed each task. Vertical black lines show the minimum and maximum number of attempts for each model and task combination.

## 5 Conclusion

In this study, we introduced a specialized RAG system designed to address the critical knowledge discovery challenge faced by large organizations with extensive historical documentation. Our approach processes over 10,000 NASA technical documents and lessons-learned reports, offering engineers quick access to verified information with direct source references, addressing a significant need for efficient knowledge retrieval in high-stakes aerospace environments.

The system demonstrated effectiveness across multiple evaluation dimensions. Retrieval experiments revealed important trade-offs: k=3, threshold=0.8 achieved highest precision (0.39) with recall (0.62), while k=10, threshold=None maximized recall (0.69) at low precision (0.07). All four evaluated models achieved consistent generation quality (BERTScore F1 0.78-0.80, groundedness scores 0.87-0.88) when provided with high-quality retrieved context, suggesting that retrieval quality, rather than model choice, determines RAG performance. The 20-participant user study validated practical utility, showing that non-technical users solved complex aerospace engineering problems with minimal attempts, indicating the system's potential to bridge knowledge gaps where valuable historical insights previously remained isolated.

Future work should test different chunking approaches such as recursive and semantic chunking, which were barely explored in this paper, and investigate whether LLMs have already been trained on NASA documentation, as this may affect evaluation validity. Additionally, implementing more advanced RAG architectures with re-ranking mechanisms, as discussed in the Related Work section, could further improve retrieval precision and answer accuracy.

required in average only 1.3 attempts to complete, demonstrating that the system effectively bridges the knowledge gap where valuable insights from past missions previously remained isolated.

These user evaluation results correspond closely with the generation evaluation findings, where all models performed very similarly when provided with high-quality retrieved context. This consistency validates the RAG approach: engineers can now quickly access verified information with direct source references. The evaluation with 20 participants completing realistic engineering scenarios provides strong evidence of the system's practical usage for NASA.

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

# A  Appendix

## A.1  System Interface Design

The RAG system was implemented as a web-based application to facilitate user evaluation and ensure consistent interaction experiences across all participants. Figure A.1 demonstrates the dual-panel interface design that integrates task management with conversational AI interaction.

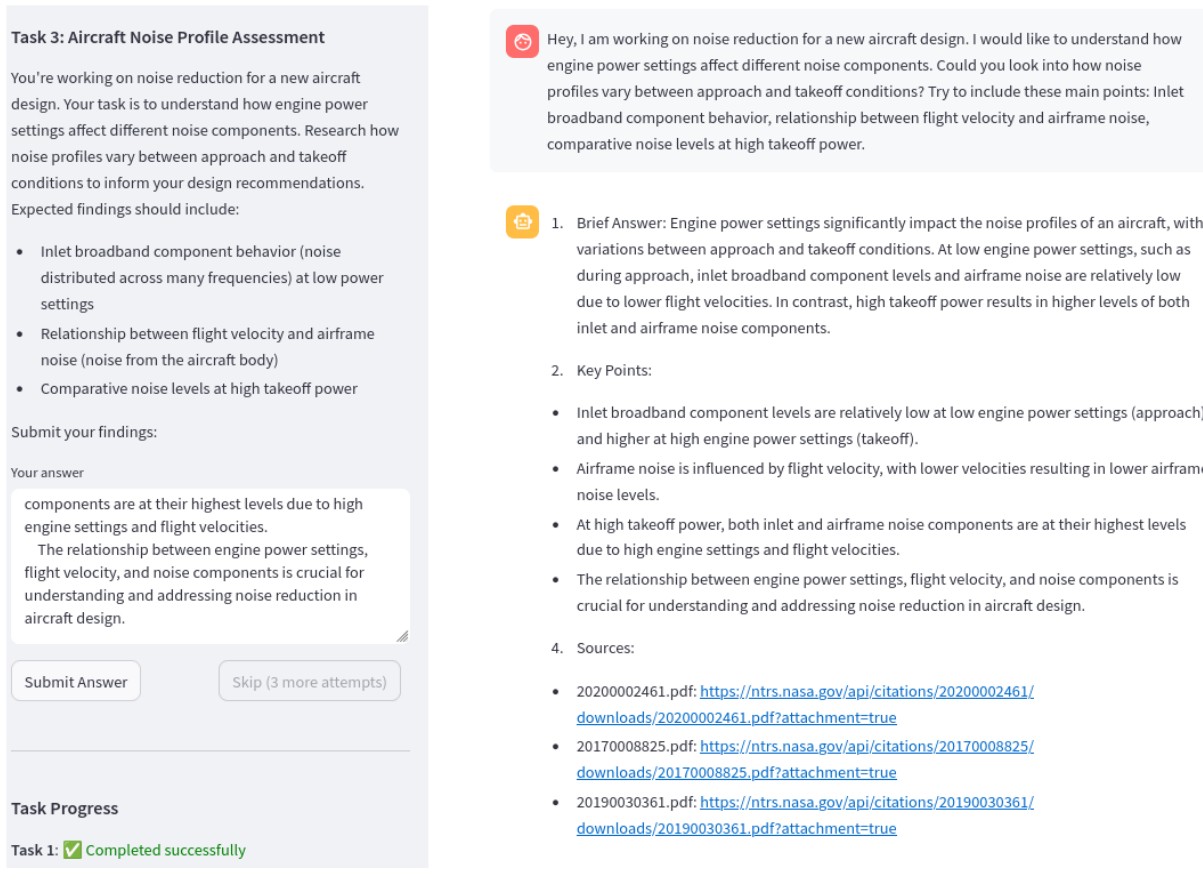

**Figure A.1.** Complete user interface of the RAG system showing the integrated evaluation environment. The left panel displays the current task description, answer submission area, and task progress tracker, while the right panel provides the interactive chat interface where users can query the AI assistant, receive contextually grounded responses, and navigate through retrieved source documents with direct links to original NASA materials.

## A.2  User Evaluation Questionnaire Details

After users completed all tasks and gained familiarity with the system, a comprehensive questionnaire was administered consisting of System Usability Scale (SUS) questions and additional questions to assess AI performance and user experience.

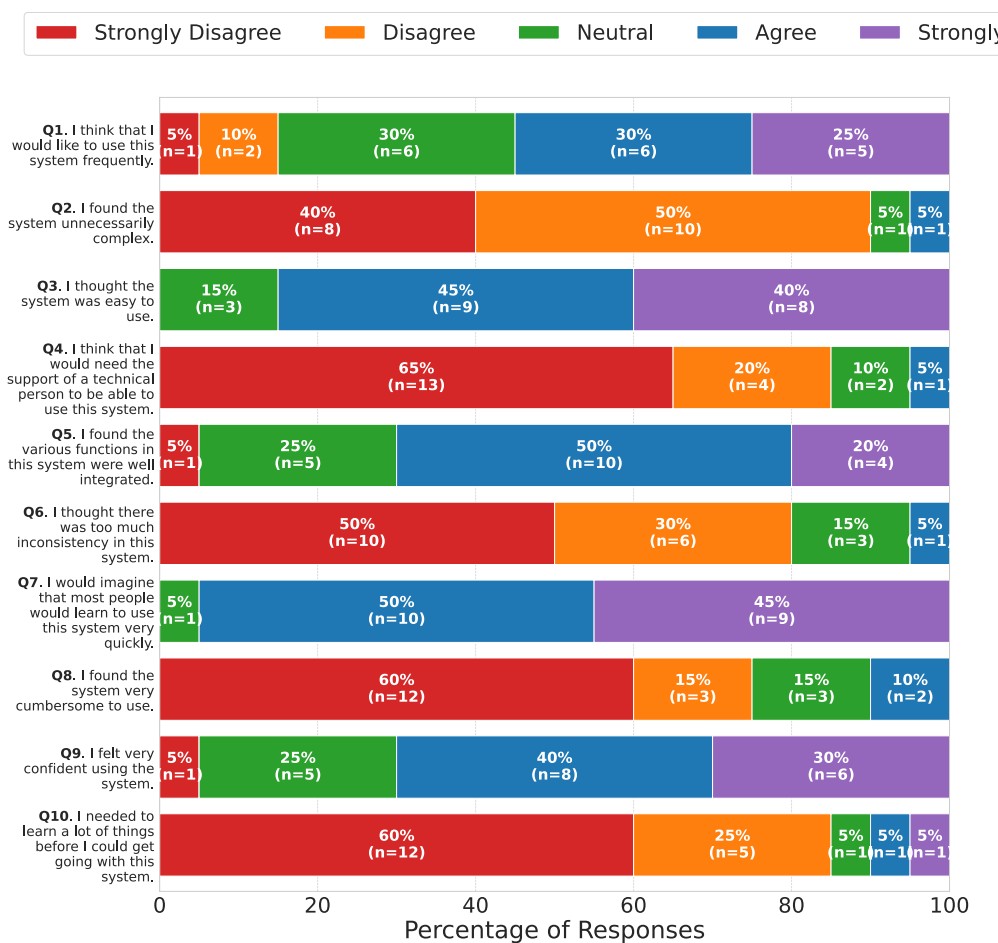

**Figure A.2.** System Usability Scale (SUS) questionnaire responses showing the distribution of user ratings across all ten SUS questions. Each horizontal bar represents one SUS question with response percentages and participant counts (n) for each rating level from Strongly Disagree to Strongly Agree.

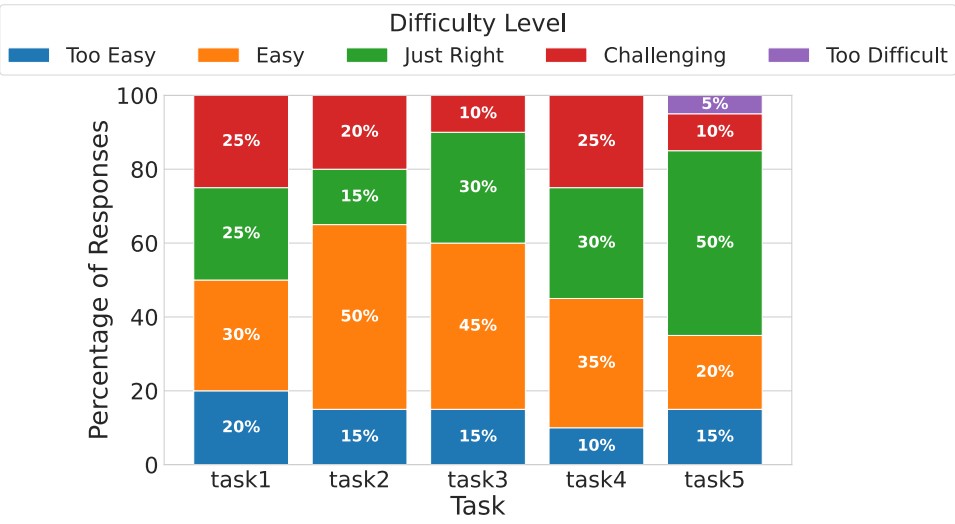

**Figure A.3.** User feedback analysis across four custom evaluation questions assessing AI system performance. The visualization shows response distributions for questions evaluating system accuracy, response quality, information usefulness, and overall satisfaction with percentage breakdowns and participant counts for each response category.

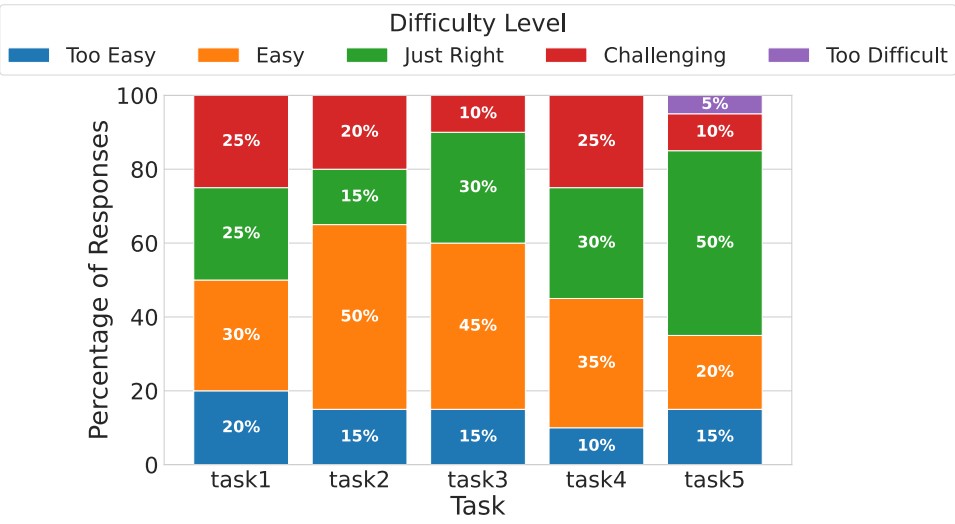

**Figure A.4.** Task difficulty assessment showing user-reported difficulty levels for each of the five evaluation tasks. The stacked bar chart displays the percentage distribution of difficulty ratings (Very Easy to Very Hard) with participant counts, providing insights into task complexity from the user perspective.

