# OpenReview forum: "RAG in the Aerospace Domain: A Retrieval, Generation, and User Evaluation for NASA Documentation"
_NLDL.org/2026/Conference — NLDL 2026 Poster_

### Official Review · Reviewer_B72E · 2025-09-24
**Interesting but unclear novelty and applicability**

**Rating:** 2
**Confidence:** 4
**Final Rating:** 4
**Final Confidence:** 4

**Summary:**

The article proposes applying retrieval-augmented generation for aerospace applications, using a vector database to retrieve relevant chunks of technical documentation which are passed to the context of the LLM answering the query. The authors scrape publicly available technical documents and postmortem incident reports from NASA, gathering a dataset of 8,143 technical reports and 1,859 "lessons learned" reports in PDF format.

The authors use the PyMuPDF4LLM library to implement section-based chunking for the dataset, segmenting the PDF documents into plaintext chunks. Embedding vectors are then generated for each chunk using the pretrained "all-MiniLM-L6-v2" embedding model by Reimers et al., and stored together with the chunks in a FAISS database.

To evaluate the system, the authors select 50 documents at random, and compose questions based on 1-3 randomly selected chunks from each document, creating 50 question-answer pairs with reference answers and relevant chunks. Models are evaluated according to lexical overlap, semantic similarity between token embeddings, and LLM-as-judge evaluations using GPT-4o-mini and the G-Eval framework.

The RAG system is then evaluated with four different models as the LLM answering queries: OpenAI's GPT-4o-mini and GPT-3.5-turbo, and Meta's LLaMa-3.3-70B-Instruct-Turbo and LLaMa-3.2-11B-Vision-Instruct. For 34 out of the 50 questions, the database returns at least one relevant chunk in the top 3 results, achieving a hit rate of 68% and 66% recall, but a top-1 precision of 23%. For the questions where a relevant chunk was identified, all models achieve lexical overlap (0.46-0.48 ROUGE-1) and high semantic similarity (0.78-0.80 BERTScore F1) with the reference answers, and high factual accuracy scores (0.88-0.90) with GPT-4o-mini as a judge.

Finally, the authors conduct a user study with 20 layman participants, tasking each with answering five different questions, requiring use of the RAG system to search the technical documentation. On average, users were able to complete these tasks in between 3 to 5 minutes each, with the majority finding the system provided relevant and helpful answers.

**Strengths:**

The authors thoroughly describe how they solved data processing issues when putting together their dataset, and other choices made during implementation. They justify their choice of chunking strategy, describe tradeoffs made during indexing, making it easier to replicate the experimental setup.

The authors have also conducted evaluations with open-weight models, and commit to publishing the dataset and the experiment code, making it easier to reproduce the results of the analysis and identify possible improvements.

The authors also corroborate the quantitative results by conducting a user study for an overall evaluation of the system, reporting both the users' performance quantitatively and their subjective experience of using the system.

**Weaknesses:**

There is little technical novelty in the article, as the RAG system consists of readily available components and previously available language models. The possible novelty is in the application to the aerospace domain: however, it is unclear whether the questions and the user study tasks reflect problems aerospace engineers regularly face, which require applying engineering principles from multiple domains, or whether the questions are limited to those which can be answered by restating facts from the documents. This concern is compounded by the user study not specifying where the users were recruited from, or whether they have aerospace engineering experience.

With the top-3 hit rate of the embedding model being 68%, and the authors reporting little variance between open-weight LLMs and commercial LLMs, the embedding model appears to be the primary bottleneck on system performance. Therefore, it is somewhat odd that only one embedding model is evaluated, and no finetuning appears to have been done on domain texts. The system performance may be improved and the article made stronger by incorporating embedding models with newer position embedding and attention techniques in the vein of ModernBERT [1]

There are also a few concerns relating to the system evaluation: the number of documents included in the context is a very important hyperparameter, but here it is fixed to a single value. Since the hit rate becomes a function of the number of documents to retrieve, the analysis would be much improved by examining the performance when this number is varied. The G-Eval framework also requires another LLM to perform the evaluations, which does not appear to be specified, raising the question of whether the OpenAI GPT models are evaluating responses from themselves.

In the user study, it also appears that at most n=12 of the 20 users completed any given task, despite all users being assigned the five tasks - is the number of reported attempts meaningful in this case?

[1] Antoun, Wissam, et al. “ModernBERT or DeBERTaV3? Examining Architecture and Data Influence on Transformer Encoder Models Performance.” Version 1, arXiv:2504.08716, arXiv, 11 Apr. 2025. arXiv.org, https://doi.org/10.48550/arXiv.2504.08716.

Suggestions which did not affect the evaluation:
* "a our" -> "our"
* LLaMa-3.2-1B-Instruct -> LLaMa-3.2-11B-Vision-Instruct (I assume the 11B model was used)

**Final Justification:**

While the article could be improved by evaluating more embedding models and hyperparameter choices, the authors have done novel work in curating the dataset and conducting a user study, and the publication of the dataset and experimental code will allow evaluation with other retrieval and language models in the future. Therefore, I am revising my rating to an accept.

**Justification:**

The technical novelty of the solution is limited, as the components described are readily available and use general pretrained language models, not models adapted to the domain text. It is also unclear whether the composition of the questions accurately simulates problems faced by aerospace engineers, a concern which is underscored by the model capabilities having little difference for the answer quality.

Additionally, the evaluation is severely limited by only examining one choice of hyperparameters for retrieval, and one embedding model for vectorization. Only quantitative measures and LLM-as-judge evaluations were used - with the small scale of the test set, expert validation of (a subset of) the answers would improve confidence in the results.

Unfortunately, while the application is interesting and the implementation is thoroughly described, these concerns make it hard to establish novelty or findings which would improve other implementations of RAG systems.

---

### Official Review · Reviewer_1rUX · 2025-09-25
**Review submission 53**

**Rating:** 4
**Confidence:** 5

**Summary:**

The submission appears as an application paper where the authors address the issue of efficiently utilizing large domain-specific knowledge bases within an organization, a very timely problem statement that has had a large interest in the industry in recent years.

The authors exemplify the problem statement by using the lessons learned documentation from NASA, an organization with a long record of solving challenging engineering tasks. The authors present Retrieval-Augmented Generation (RAG) as a viable solution to interact with large knowledge sources, a technique that is popular in both industry and that has had immense research focus in the past 2-3 years. The paper then provides a brief and superficial summary of previous work within RAG and RAG evaluation.

In section 3, the authors introduce their system design. The model system design is a basic bear-bone RAG system with a dense retriever and an LLM for text completion. The system thus does not utilize common RAG techniques such as query rewriting or reranking, nor are these techniques mentioned in the "related work" or "method" sections.
The data collection process involves several sources and a substantial amount of documents; thus, it appears thorough and well-conducted. The technical implementations and design choices are then presented and discussed properly.

The authors then present their evaluation framework, which is three-sided: 1) The retrieval component is first evaluated using standard metrics from the IR field (Recall, NDCG, MRR, ...). 2) The generation is then evaluated using a vast set of metrics, including trained metrics (BERTscore), lexical metrics (ROUGE), and LLM-as-a-judge metrics (GEval and GPT-based). 3) A user study where a small number of users with limited technical knowledge are involved in the evaluation.

The automatic evaluation is done on 50 tasks, while the human evaluation is conducted on 5 tasks. The authors have included examples from the test dataset that showcase the data structure, thus providing the reader with a better understanding of the underlying task.

The result section presents the results from all three evaluations in a systematic and organized manner. Tables and plots are included to provide insight for the reader. The results are discussed on the fly, rather than in a separate discussion section, but this is done properly. For the retrieval evaluation, the authors only present metrics for one implementation of the system, although alternative implementations, like chunking-strategy, are discussed in the methodology. The results show a recall of 66%; thus, on many occasions, there will not be any relevant documents retrieved by the system. This is part of the results that should be discussed. For the generation evaluation, the authors have tested four LLMs across 14 metrics. This is a comprehensive evaluation, but the authors only discuss parts of the findings. Their discussion of the "relevance" metrics, which are reported by GPT-4 and by G-Eval, provides useful insight into the evaluation of RAG systems, a field that has many open questions. Nonetheless, the authors do not go in-depth on what causes this metric to differ between the two sources.  For the user testing, the authors report completion time and the number of attempts users needed to complete the 5 separate tasks. Results are reported for both the GPT-4o-mini model and the LLaMA.3.3-70B-Instruct-Turbo model, which provide a useful comparison between open-source models and proprietary models.

The authors then summarize the contribution in the conclusion section.

**Strengths:**

The paper addresses the use of RAG for domain-specific applications, a very timely task that has received significant attention in both academic research and industry. The paper is well organized and easy to follow when reading. The system design is well presented, along with the data collection and evaluation setup. The use of over 10.000 NASA documents provides a strong data foundation for the project, and this data is hopefully shared afterwards. The usage of both open source and proprietary models provides an insightful comparison that lifts the quality of the submission. The inclusion of a user study adds significant academic value as RAG systems are designed for end-users, while many papers do not focus on the actual real-world applications of RAG systems, but rather their automatic evaluation metrics. The discussion on how "relevance" is reported by both G-Eval and GPT-4 provides insight into the evaluation of RAG models and generative AI, which have many open questions.

All in all, the contribution showcases how RAG can be utilized in a new domain, while providing useful insight into the difference between open source and proprietary models, and on RAG evaluation.

**Weaknesses:**

Although showing promising results, the RAG pipeline is primitive and potentially limits the results. The authors claim the retrieval results to be good. Although they are significantly better than random selection, the results might not be strong enough for a real-world application. The reported recall of 66 % would indicate that there are many occasions where the IR component does not find the relevant documents, thus the end-user is not provided with a grounded result, which, for aerospace applications, can have fatal consequences. The authors should address this, either by arguing why 66 % is sufficient, or by conducting extended experiments for the IR component (varying "k", adding a re-ranker, performing query re-writing, ...).

The data foundation is only 50 items; this can be seen as a low amount. The authors should address whether this number of test items is sufficient.

The choice of metrics for evaluating the generative part is not justified well enough. The strength and relevance of the metrics should be grounded in the application. The author should intend to answer "Why is metric X relevant for a RAG system used by NASA?" and "What does a value of Y for metric X mean for an end user?". This would help justify the metrics and the comparison of the various models.

The GPT-4 model is used both for generation and for evaluation, thus providing potential bias. This should be addressed.

In Table 3, please highlight the top results in bold font.

The insight into how "relevance" is reported differently by both G-Eval and GPT-4 due to their underlying difference in prompt design (single-prompt vs multi-prompt) is an interesting finding in the paper. Nonetheless, the "why" is not fully explained; thus, the paper would benefit from a more in-depth discussion on this matter.

The "related work" section could benefit from a deeper insight into RAG designs with more complex architectures than the vanilla setup of a dense retriever and an LLM.

The title hints at a comprehensive system design, which should include more testing of design options. I suggest removing the word "comprehensive" from the title to better reflect the content of the submission.

**Justification:**

The submission is clearly an application paper addressing a relevant topic. The paper is well structured and provides new insight, especially within its user study, comparison between open source and proprietary models, and evaluation of RAG systems.


Nevertheless, the paper has some clear drawbacks. The related works section is quite weak, especially related to the design of RAG systems, which is also reflected in the methodology section. This makes the contribution less comprehensive, thus not providing insight into system design. Also, there are several weaknesses with the results and evaluation setup that are not addressed by the authors.

---

### Official Review · Reviewer_rCXu · 2025-10-04
**In this paper, the authors present a web-based RAG system using documents from NASA, and investigate the effectiveness of the RAG system for both technical and non-techniqual users.**

**Rating:** 2
**Confidence:** 4

**Summary:**

RAG is a useful technique in many LLM-based applications. In this paper, the authors present a web-based RAG system using documents from NASA, and investigate the effectiveness of the RAG system for both technical and non-techniqual users.

**Strengths:**

I think it is helpful to build domain-specific RAG systems and analyze the effectiveness and challenges of RAG in different domains. And such an investigation can provide valuable insights and references for future studies.

**Weaknesses:**

However, my main concerns about this paper are as follows:
1. The representativeness of the constructed RAG system. Currently, there are already many standard solutions for RAG. The RAG framework in this paper contains numerous ad hoc design choices (e.g., the document chunking and embedding choices), which limit the representativeness of their proposed solution and the conclusions.
2. The authors should analyze the challenges of RAG in the aerospace domain, and whether these challenges are domain-specific or universal in different domains. As a result, it is difficult to assess whether the conclusions in this paper are general in different RAG systems.
3. Most conclusions and discussions in this paper are just description of experimental results, I think more insightful analysis are needed for valuable conclusions.

**Justification:**

Overall, I think the construction and the analysis of domain-specific RAG is valuable, but the representativeness of the RAG system proposed in this paper is limited, and the conclusions are not insightful .

---

> ### Author Rebuttal · Authors · 2025-10-20
>
> We thank all the reviewers for their thoughtful feedback.
>
> **Comment to all the reviewers:**
> As stated, this paper’s primary contribution is applied rather than architectural. It focuses on evaluating the practical deployment of RAG in a domain specific context, aerospace knowledge retrieval, which remains largely unexplored in current literature. We believe that domain adaptation, data gathering, evaluation design, and user study insights are valuable contributions even when using existing RAG components especially since all of the work will be publicly available.
>
> We acknowledge that the experimental evaluation of the RAG component was insufficient to fully support the conclusions and demonstrate its contribution. Thus, in the camera-ready version, we will provide additional experimental results with varying _k_-threshold values and different numbers of retrieved documents.
>
> **Comment to the 1. Reviewer (rCXu):**
> We have experimented with additional setups, such as different chunking methods, however, we aimed on describing only the most successful setup and application side of the described use case, as our focus is rather than fine-tuning the RAG system on developing and testing a proof-of-concept application.
>
> **Comment to the 2. Reviewer (1rUX):**
> We acknowledge that a recall of 66% may seem modest; however, this value is comparable to retrieval performance reported in related work on dense retrievers, such as Dense Passage Retrieval (Karpukhin et al. [https://arxiv.org/abs/2004.04906](https://arxiv.org/abs/2004.04906 "https://arxiv.org/abs/2004.04906")), which achieved top-5 retrieval accuracy around 68–78% on open-domain datasets. Considering that our dataset consists of highly specialized NASA technical reports with limited vocabulary overlap, the obtained recall can be considered within the expected range.
>
> Regarding the comment about evaluating on 50 queries, prior literature has shown that this number is sufficient ([https://nlp.stanford.edu/IR-book/pdf/08eval.pdf](https://nlp.stanford.edu/IR-book/pdf/08eval.pdf "https://nlp.stanford.edu/IR-book/pdf/08eval.pdf")) to distinguish quality of the systems and 50 queries are thus used for evaluating Information Retrieval systems as a standard (e.g. in the CLEF or TREC campaigns).
>
> More in-depth discussion of the selected metrics, with respect to the task and a comparison of the single-prompt and multi-prompt will be included in the camera-ready version.
> Related work section will be expanded in the camera-ready version of the paper. We will also remove the word 'Comprehensive' from the title.
>
> **Comment to the 3. Reviewer (B72E):**
> Regarding the domain relevance and question design, this system is based on our in-person discussions with NASA engineers and this problem was attempted to be solved at NASA Risky Business challenge [https://www.freelancer.com/contest/Risky-Space-Business-NASA-AI-Risk-Prediction-Challenge-2008562](https://www.freelancer.com/contest/Risky-Space-Business-NASA-AI-Risk-Prediction-Challenge-2008562 "https://www.freelancer.com/contest/Risky-Space-Business-NASA-AI-Risk-Prediction-Challenge-2008562") showing its necessity.
>
> We agree that newer models, such as ModernBERT are expected to perform better and further improve the scores. However we choose miniLM as a lightweight model, with still robust and competitive performance. As mentioned, our focus was rather than on fine-tuning of the application, on building a proof-of-concept application from scratch. This included collecting in-domain data and human evaluation, including testing the solution in real-time, in which miniLM proved to be efficient.
>
> Regarding the user study, participants were randomly divided between the two LLM conditions (GPT and LLaMA) for each task, ensuring that both models were evaluated on the same set of five tasks. Thus, while only _n = 12_ users may appear for a given model on a specific task, the remaining _n = 8_ users completed the same task with the other model. In total, all 20 participants completed every task, but the results were split across models to enable a balanced comparison.

---

### Meta-Review · Area_Chair_1nia · 2025-11-01

**Recommendation:** Accept (Poster)
**Confidence:** 2

**Metareview:**

The paper presents the implementation and evaluation of a RAG query system based on large language models to retrieve NASA documents.

While the tools and methods used for the retrieval are existing and no specific methodological advance is proposed, some of the reviewers acknowledge the contribution to the user case study and the fact that the data and the work will be publicly available, therefore allowing for improvement on the current limited retrieval performance.

pros:
1. Evaluation of a RAG system in a specific application domain, NASA technical documents
2. The work will be publicly available
3. Represents an example use case study in this domain
4. The authors agree on the extension of the evaluation as a function of the document fragments used

cons:
1. No application-specific methodology proposed
2. general lack of technical novelty
3. Limited insight into the limitations of the current solution
4. limited evaluation

Overall, it is difficult to suggest either an acceptance or a rejection.

Indeed, while there is little technical contribution, the application and the data collected could be of inspiration for future advancement.

I am therefore suggesting accepting, but with quite low confidence.

---

### Decision · Program_Chairs · 2025-11-05

**Decision:**

Accept (Poster)

**Comment:**

We recommend a poster presentation given the AC and reviewers recommendations.